

# Effects of different exercise types on vascular endothelial function in individuals with abnormal glycaemic control: a systematic review and network meta-analysis

Zongxiang Li[1], Shengyao Luo[1], Xuebing Bai[1], Lu Huang[1], Hongyan Guo[1], Song Chen[1] and Dan Wang[2]

[1] Faculty of Physical Education and Art, Jiangxi University of Science and Technology, Ganzhou, China
[2] Provincial University Key Laboratory of Sport and Health Science, School of Physical Education and Sport Sciences, Fujian Normal University, Fuzhou, China

## ABSTRACT

**Background**. Brachial artery flow-mediated dilation (FMD) is a key marker of endothelial function, often impaired in individuals with abnormal glycemic control. While exercise has been shown to improve brachial artery FMD, the relative efficacy of different exercise modalities remains unclear. This study employed a network meta-analysis (NMA) to compare the effects of various types of exercise on FMD.

**Methods**. A comprehensive search of PubMed, Embase, Cochrane, Web of Science, and EBSCO databases identified randomized controlled trials evaluating the effects of exercise on brachial artery FMD up to January 2025. Two independent reviewers screened studies, extracted data, and assessed risk of bias. Eligible studies were assessed for bias using version 2 of the Cochrane Risk of Bias tool. Stata 16.0 was used for the NMA.

**Results**. Seventeen studies with 797 participants (prediabetes: 76; T2DM: 721) were included. Aerobic interval exercise (AIE) significantly improved FMD (MD = 2.23%, 95% CI [1.0 9%–3.37%], $P < 0.05$), followed by mind-body exercise (MBE) (MD = 1.97%, 95% CI [0.60%–3.33%], $P < 0.05$). Combined exercise (CE) (MD = 1.17%, 95% CI [0.13%–2.21%], $P < 0.05$) and aerobic continuous exercise (ACE) (MD = 1.20%, 95% CI [0.52%–1.87%], $P < 0.05$) also showed significant improvements. SUCRA values indicated that AIE (89.0) and MBE (80.1) were the most effective in improving FMD, followed by CE (51.0), ACE (50.9), and resistance exercise (RE) (20.1), all outperforming the control group (SUCRA = 9.2).

**Conclusion**. AIE was the most effective modality for improving FMD, with MBE serving as a viable alternative for individuals with lower fitness or cardiovascular concerns. CE and ACE also provided benefits, while RE was less effective. Future studies should focus on long-term outcomes and personalized exercise strategies.

Corresponding author
Zongxiang Li, 1179764031@qq.com

## INTRODUCTION

Diabetes is one of the most prevalent chronic diseases worldwide, exerting a significant impact on the well-being of individuals and societies (*Ong et al., 2023*). In 2019, approximately 463 million people (9.3% of the global population) were living with diabetes, and this number is projected to exceed 578 million (10.2% of the global population) by 2030 (*Saeedi et al., 2019*). Furthermore, according to the *2021 Diabetes Atlas* released by the International Diabetes Federation, around 541 million adults worldwide are estimated to have prediabetes, representing a prevalence rate of approximately 10.6% (*Federation, 2021*). Among individuals with prediabetes at the age of 45, about three-quarters are likely to develop type 2 diabetes mellitus (T2DM) during their lifetime (*Ligthart et al., 2016*). Therefore, it is crucial to pay adequate attention to and intervene in abnormal glycemic control, even in its early stages.

Endothelial cells, which form a monolayer covering the inner wall of blood vessels, play a crucial role in maintaining vascular homeostasis (*Sun et al., 2020*). However, in pathological states, endothelial cell function is impaired, manifesting as reduced vasodilation capacity, enhanced pro-thrombotic and pro-inflammatory states, collectively referred to as endothelial dysfunction (*Avogaro et al., 2011*; *Theofilis et al., 2021*). This phenomenon is not only an early marker of atherosclerosis (*Fetterman et al., 2016*), but also a significant pathological feature of prediabetes and T2DM (*Avogaro et al., 2011*; *Lamprou et al., 2023*). Studies have shown that individuals with prediabetes already exhibit endothelial dysfunction, with the severity of dysfunction positively correlated with insulin resistance and blood glucose levels (*Liang et al., 2021*; *Wasserman, Wang & Brown, 2018*), and it can independently predict the risk of diabetes progression (*Meigs et al., 2004*). In T2DM patients, endothelial dysfunction is more pronounced and is closely related to the duration of the disease and the level of blood glucose control (*Naka et al., 2012*; *Shi & Vanhoutte, 2017*), playing a key role in the development of diabetes-related microvascular and macrovascular complications (*Shi & Vanhoutte, 2017*). Therefore, improving endothelial function is considered an important target for the prevention and treatment of abnormal glucose control and its complications. Currently, methods to assess endothelial function primarily include flow-mediated dilation (FMD), peripheral arterial tonometry (PAT), and circulation endothelial cell detection, among others (*Poredos & Jezovnik, 2013*). Among them, FMD, which measures the brachial artery's response to increased blood flow *via* ultrasound, has become the gold standard in clinical research and practice for assessing large vessel endothelial function due to its non-invasive nature and good reproducibility (*Alley et al., 2014*; *Thijssen et al., 2019*). Multiple pharmacological interventions have been shown to effectively improve FMD in patients with T2DM, including statins (*Mokgalaboni, Dludla & Nkambule, 2022*; *Zhang et al., 2012*), sodium-glucose cotransporter 2 inhibitors (*Wei et al., 2022*), and phosphodiesterase inhibitors (*Santi et al., 2015*). However, these treatments may exhibit inter-individual variability in efficacy, and long-term use may lead to adverse effects, potentially reducing patient adherence (*Alanezi, 2025*; *Alshahrani & Almalki, 2024*; *Tan et al., 2019*).

Exercise, diet, and medication are the main interventions for managing T2DM and prediabetes (*Samson et al., 2023*; *Stevens et al., 2015*). Among these, exercise is regarded as one of the most promising non-pharmacological treatments due to its cost-effectiveness, ease of implementation, and absence of drug-related side effects. Research indicates that exercise can significantly improve cardiovascular function (*Pinckard, Baskin & Stanford, 2019*), inflammation levels (*Pedersen, 2017*), cognitive function (*Gu et al., 2019*), and metabolic health (*Zhao et al., 2024*) in individuals with T2DM and prediabetes. An umbrella reviews have shown that exercise significantly improves FMD in different populations (*Shivgulam et al., 2023*), underscoring its importance in enhancing endothelial function. As consensus on exercise as a treatment approach gradually forms (*Sigal et al., 2006*), its potential benefits for FMD have attracted widespread attention (*Abdi, Tadibi & Sheikholeslami-Vatani, 2021*; *Afousi et al., 2018*; *Choi et al., 2012*; *Cox et al., 2024*; *Davoodi et al., 2022*; *Desch et al., 2010*; *Gainey et al., 2016*; *Gibbs et al., 2012*; *Ku et al., 2009*; *Kwon et al., 2011*; *Li, 2009*; *Liu, Li & Liu, 2007*; *Mitranun et al., 2014*; *Okada et al., 2010*; *Ploydang et al., 2023*; *Rech et al., 2019*; *Wycherley et al., 2008*). However, existing studies show significant variability in exercise type selection and effect evaluation. Aerobic exercise, while improving FMD, typically requires lower intensity and longer duration, which may present adherence issues for individuals with obesity-related diabetes (*Forhan et al., 2013*). Therefore, resistance exercise and aerobic interval exercise may be more feasible alternatives (*Francois et al., 2016*; *Kourek et al., 2023*). Currently, comparative studies on the effects of different exercise types on FMD remain limited, with inconsistent results. Furthermore, meta-analyses have certain limitations in terms of study inclusion. For instance, some meta-analyses have included only individuals with T2DM while excluding prediabetic populations, potentially limiting the generalizability of their findings (*Lee et al., 2018*; *Way et al., 2016*). Additionally, some studies have assessed endothelial function using FMD measurements from different vascular sites (*e.g.*, brachial and femoral arteries), and different vascular sites may have varying responses to endothelial function, which could account for inconsistencies in results (*Chen et al., 2023*; *Qiu et al., 2024*; *Qiu et al., 2018*).

Therefore, this study aims to explore the differential effects of various types of exercise (continuous aerobic, interval training, resistance exercise, *etc.*) on brachial artery FMD in individuals with abnormal glucose control, and rank the interventions according to their effectiveness, through a systematic review and network meta-analysis (NMA). This study is intended for clinical researchers, exercise physiologists, endocrinologists, and public health professionals seeking evidence-based guidance on optimizing exercise interventions for improving vascular health in individuals with impaired glycaemic control.

## METHODS

### Registration

This systematic review and NMA were performed in accordance with the Preferred Reporting Items for Systematic Reviews and Meta-Analyses (PRISMA) guidelines (*Hutton et al., 2015*). The study protocol has been registered in the International Prospective Register of Systematic Reviews under ID: CRD42025643867.

## Literature search strategy

A comprehensive literature search was performed across multiple databases, including PubMed, Web of Science, Cochrane Library, Embase, and EBSCO, to identify randomized controlled trial (RCT) examining the impact of exercise on brachial artery FMD in individuals with impaired glycemic control, specifically those with prediabetes or T2DM. The search period spanned from the inception of each database to January 2025. In addition, Google Scholar was searched, and the reference lists of pertinent reviews and meta-analyses were manually screened for additional relevant studies. The search utilized Medical Subject Headings (MeSH) terms in PubMed and Cochrane Library, as well as Emtree terms in Embase. The strategy involved the use of key search terms, including "Exercise" OR "Type 2 Diabetes Mellitus" AND "Endothelial Function." Detailed information on the full search strategy can be found in Appendix S1.

## Eligibility criteria

Eligibility criteria were established based on the Population, Intervention, Comparator, Outcomes, and Study (PICOS) framework (*Amir-Behghadami & Janati, 2020*).

## Inclusion criteria

1. Population: Adults aged 18 years or older with abnormal glycemic control, specifically those with prediabetes or T2DM. Prediabetes was defined as meeting at least one of the following criteria for baseline glucose parameters: fasting glucose between 5.6 and 6.9 mmol/L, 2-hour glucose between 7.8 and 11.0 mmol/L, or HbA1c between 5.7% and 6.4% (*American Diabetes Association Professional Practice Committee, 2022*). T2DM was defined by at least one of the following criteria: fasting glucose ≥7.0 mmol/L, 2-hour glucose ≥11.1 mmol/L, HbA1c ≥6.5%, or random plasma glucose ≥11.1 mmol/L (*American Diabetes Association Professional Practice Committee, 2022*).

2. Intervention: Exercise, defined as planned, structured, and repetitive physical activity aimed at improving or maintaining physical fitness (*Caspersen, Powell & Christenson, 1985*). This includes interventions such as aerobic continuous exercise (ACE), aerobic interval exercise (AIE), resistance exercise (RE), mind-body exercise (MBE), and combined exercise (CE), as outlined in Table 1 (*Martínez-Vizcaíno et al., 2022*). The experimental group underwent structured exercise training for at least 6 weeks.

3. Comparator: Comparisons could be made between the exercise group and a control group, or among different exercise modalities. The control group adhered to their routine physical activities, received health education, and performed stretching exercises, or maintained their usual daily lifestyle, while the experimental group underwent a structured exercise training program in addition to the control regimen.

4. The sole outcome measure in this review was FMD, which serves as a marker of endothelial function and vascular health. Only studies that assessed endothelial function through brachial artery FMD were included. The FMD measurement was performed by calculating the absolute diameter change (peak diameter–baseline diameter), with subsequent conversion to percentage values using the standard equation: %FMD = (absolute FMD/baseline diameter) ×100.

**Table 1  Definition of exercise types.**

| Types of exercise | Abbreviation | Definition |
|---|---|---|
| Aerobic continuous exercise | ACE | This type of exercise involves sustained activities that elevate the heart rate and enhance energy expenditure at a moderate, consistent intensity. Examples include walking, jogging, running, and cycling. |
| Aerobic interval exercise | AIE | Aerobic interval training consists of alternating bursts of high-intensity efforts with periods of lower intensity or rest for recovery. These intense intervals typically push the body to its maximum capacity, improving aerobic fitness. It encompasses methods such as high-intensity interval training (HIIT). |
| Resistance exercise | RE | Resistance training targets muscle strength by using external resistance, such as dumbbells, resistance bands, or machines, to enhance power. |
| Mind-body exercise | MBE | Mind-body exercises combine physical movement with mental focus, fostering both physical and psychological well-being. These interventions include practices such as Tai Chi, yoga, Baduanjin, and Wuqinxi, among others. |
| Combined exercise | CE | Combined exercise programs incorporate various forms of physical activity, blending different types of exercises such as aerobic and resistance training. |
| Control group | CON | no exercise or light stretching |

5. Study design: Only RCT, including cluster-randomized and crossover designs, were included.

## Exclusion criteria

Studies were excluded if they met any of the following criteria:

1. Focused on pregnant women, individuals with type 1 diabetes, or patients with acute medical conditions (*e.g.*, those presenting in emergency departments).

2. Were duplicates, literature reviews, letters to the editor, conference abstracts, animal studies, or low-quality literature.

3. Had unavailable full-text articles or incomplete study data.

4. Did not report vascular function outcomes relevant to this review.

5. Combined exercise interventions with diet, insulin, or other pharmacological treatments.

## Study selection and data extraction

The process of screening titles, abstracts, and full texts was carried out independently by two authors (Zhongxiang Li and Shengyao Luo), following the established inclusion and exclusion criteria. Any discrepancies were resolved through discussions with Dan Wang.

The data extracted from the included studies comprised several key pieces of information: the first author, year of publication, and participant demographics (including the number of individuals in both experimental and control groups, age, and gender). Information on the exercise intervention was also recorded, which included details such as the type of exercise, its intensity, duration, frequency, and session length. Brachial artery FMD

outcomes were the primary measure of interest. In instances where certain data were missing, the corresponding authors were contacted *via* email in an effort to retrieve the required information.

### Risk of bias assessment

The risk of bias was evaluated using version 2 of the Cochrane Risk of Bias (RoB 2) tool, which introduces several important advancements for assessing bias in randomized trials (*Sterne et al., 2019*). This tool encompasses five key domains: (1) the randomization process, (2) deviations from intended interventions, particularly allocation concealment, (3) handling of missing outcome data, (4) outcome measurement, and (5) selection of reported results. Using robvis tool to create risk-of-bias plots (*McGuinness & Higgins, 2021*).

### Data synthesis and statistical analysis

The effect was estimated in this NMA by combining the changes observed before and after the intervention in both the experimental and control groups. The calculation method of the standard deviation (SD) of the change was based on the formula provided by the Cochrane Handbook for Systematic Reviews of Interventions (version 6.3) (*Higgins et al., 2022*) (the formula is: $SD_{change} = \sqrt{SD_{baseline}^2 + SD_{final}^2 - (2 \times Corr \times SD_{baseline} \times SD_{final})}$). In this formulate, a correlation coefficient (Corr) of 0.5 was assumed, which is commonly used when the actual correlation is not reported in the original studies (*Follmann et al., 1992*). The sensitivity of results to this assumption was tested by varying Corr between 0.3 and 0.7 in a sensitivity analysis (see Appendix S2).

Data were analyzed using META through Stata 16.0 software (StataCorp, College Station, TX, USA), with outcome indicators treated as continuous variables. For brachial artery FMD, mean difference (MD) and their corresponding 95% confidence intervals (CI) were calculated, with baseline adjustments made at $\alpha = 0.05$. Heterogeneity was assessed using the Q-test and $I^2$ statistic. Publication bias was assessed using Egger's test and funnel plots (including a network funnel plot for NMA), with symmetry as the evaluation criterion. A random-effects multivariate NMA was conducted using a frequency framework with STATA 16.0 software (*Salanti, 2012*), following the most recent PRISMA NMA guidelines (*Hutton et al., 2015*). A network diagram was created to visually depict the relationships between different intervention strategies. The lines connecting the nodes represent direct comparisons between the interventions, with the size of the nodes and the thickness of the lines reflecting the number of studies. Furthermore, a network contribution plot was generated to evaluate the contributions of each direct comparison.

Transitivity assumptions were evaluated through the inclusion criteria of individual studies, by assessing whether all participants in the network could be randomly assigned to any intervention, and by applying consistency models (*Rücker & Schwarzer, 2015*). As a key assumption in NMA, transitivity asserts that indirect comparisons are valid approximations of unobserved direct comparisons (*Salanti, 2012*) and that effect modifiers are uniformly distributed across studies (*Jansen & Naci, 2013*). To evaluate consistency in each closed loop, inconsistency factors (IFs) with 95% CIs were calculated, with consistency indicated when the lower limit of the 95% CI equals 0 (*Chaimani et al., 2013*). The inconsistency

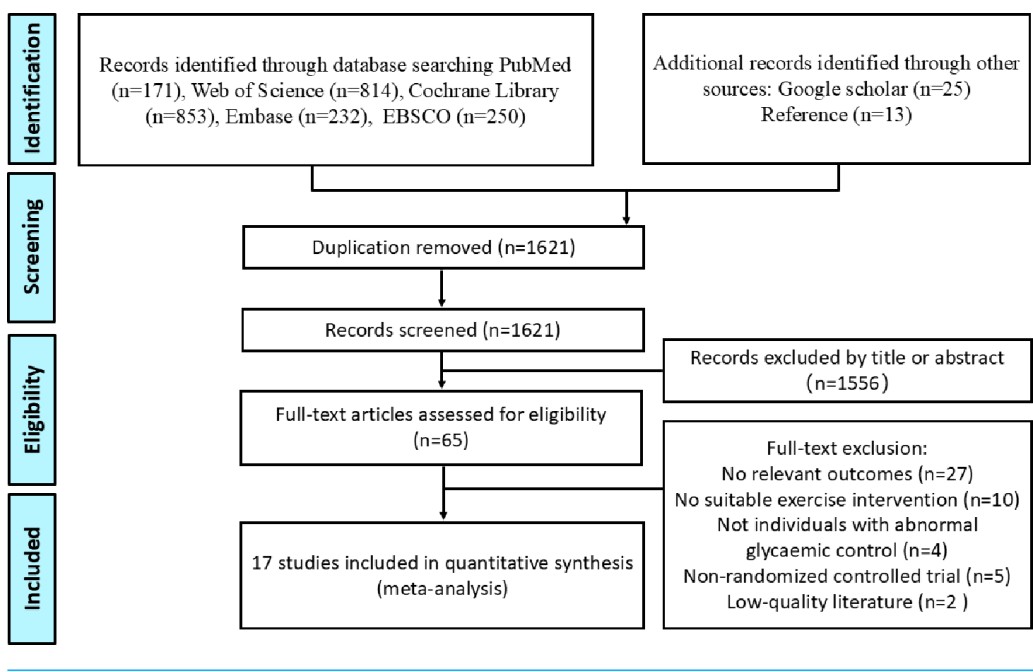

**Figure 1** Flowchart of the literature-retrieval protocol.

model was applied to test for inconsistency, while the consistency model was used when no significant inconsistency was detected ($P > 0.05$) (*Shim et al., 2017*). Node-splitting analysis was conducted to assess local inconsistency, and the results were deemed reliable with $P > 0.05$. The surface under the cumulative ranking curve (SUCRA) was applied to rank and compare the effectiveness of various exercise interventions (*Salanti, Ades & Ioannidis, 2011*). SUCRA values range from 0 to 100, with higher values indicating more favorable effects of exercise interventions (*Mbuagbaw et al., 2017*).

## RESULT

### Literature selection

Figure 1 outlines the process of literature search and selection. A total of 2,320 articles were initially retrieved from five databases, with an additional 25 articles identified through Google Scholar. In addition, 13 relevant articles were sourced from reference lists. After removing 737 duplicates, 1,621 articles remained for the screening phase. Of these, 1,556 were excluded based on the review of titles and abstracts. Following a full-text review, 48 additional articles were excluded, resulting in 17 articles that were ultimately included in the NMA.

### Literature characteristics

The characteristics of the studies included in this research are detailed in Appendix S3. A total of 797 individuals with abnormal blood glucose control were included in this study (Prediabetes: 76; T2DM: 721). The average age ranged from 33 to 70.5 years. Four studies (*Abdi, Tadibi & Sheikholeslami-Vatani, 2021*; *Choi et al., 2012*; *Ku et al., 2009*; *Kwon et al.,*

*2011*) exclusively included female T2DM patients, while *Liu, Li & Liu (2007)* did not specify the gender of the participants. In the study by *Gainey et al. (2016)*, 81.8%–83.3% of the participants were female, whereas in *Desch et al. (2010)*, only 22%–34% of the participants were female. In the remaining studies, the gender ratio of male to female participants was approximately 1:1. Based on the type of exercise intervention, 186 individuals were assigned to ACE, 63 to AIE, 30 to RE, 109 to CE, 52 to MBE, and 357 to control group (CON). The exercise intensity in the ACE group was relatively consistent, with a range of approximately 50%–75% HRmax. The intensity in the AIE group also showed little variation, with exercise intensity ranging from 80%–95% HRmax, and active rest ranging from 50%–60% HRmax. The exercise intensity in the RE, CE, and MBE groups was similarly consistent. With the exception of *Rech et al. (2019)* which involved light stretching, no other studies in the control group included exercise interventions. The duration of exercise interventions in all 17 studies was ≥8 weeks, with over half (70.59%) of the studies having an intervention duration of 12 weeks. All 17 studies included supervised training.

## Risk of bias assessment results

The results of the bias risk assessment are provided in Appendix S4. In the D1 (randomization process) domain, 14 studies were assessed as having a low risk of bias. However, three studies described the use of random allocation methods but failed to provide specific details on the random sequence generation process. In the D2 (deviation from intended interventions) domain, all studies adhered to the intended interventions, resulting in a low risk of deviation. In the D3 (missing outcome data) domain, six studies had unclear reporting of missing data, indicating a potential risk of bias. Additionally, a dropout rate exceeding 20% was considered high risk, and three studies were classified as high risk based on this criterion (*Chen et al., 2024*). In the D4 (outcome measurement) domain, all studies employed standardized measurement methods, resulting in a low risk of bias. Finally, in the D5 (selection of reported results) domain, 12 studies raised some concerns due to the lack of mention of RCT protocol registration.

## Network meta-analysis

Figure 2 illustrates the NMA plots of eligible studies assessing the impact of various exercise categories on brachial artery FMD. The size of the nodes corresponds to the sample size within each exercise category, while the thickness of the lines connecting exercise types reflects the number of studies evaluating that particular comparison. The most frequently utilized intervention was ACE, whereas MBE was the least common.

Appendix S5 outlines the contributions of direct and indirect comparisons to the NMA, as well as the number of studies for each direct comparison. The consistency of brachial artery FMD was evaluated using loop-specific heterogeneity estimates, the inconsistency model, and node-splitting analysis (Appendix S6). Loop-specific heterogeneity estimates revealed high consistency within each closed loop for brachial artery FMD. The inconsistency model showed no significant discrepancies between direct and indirect comparisons ($P > 0.05$). Furthermore, node-splitting analysis indicated no
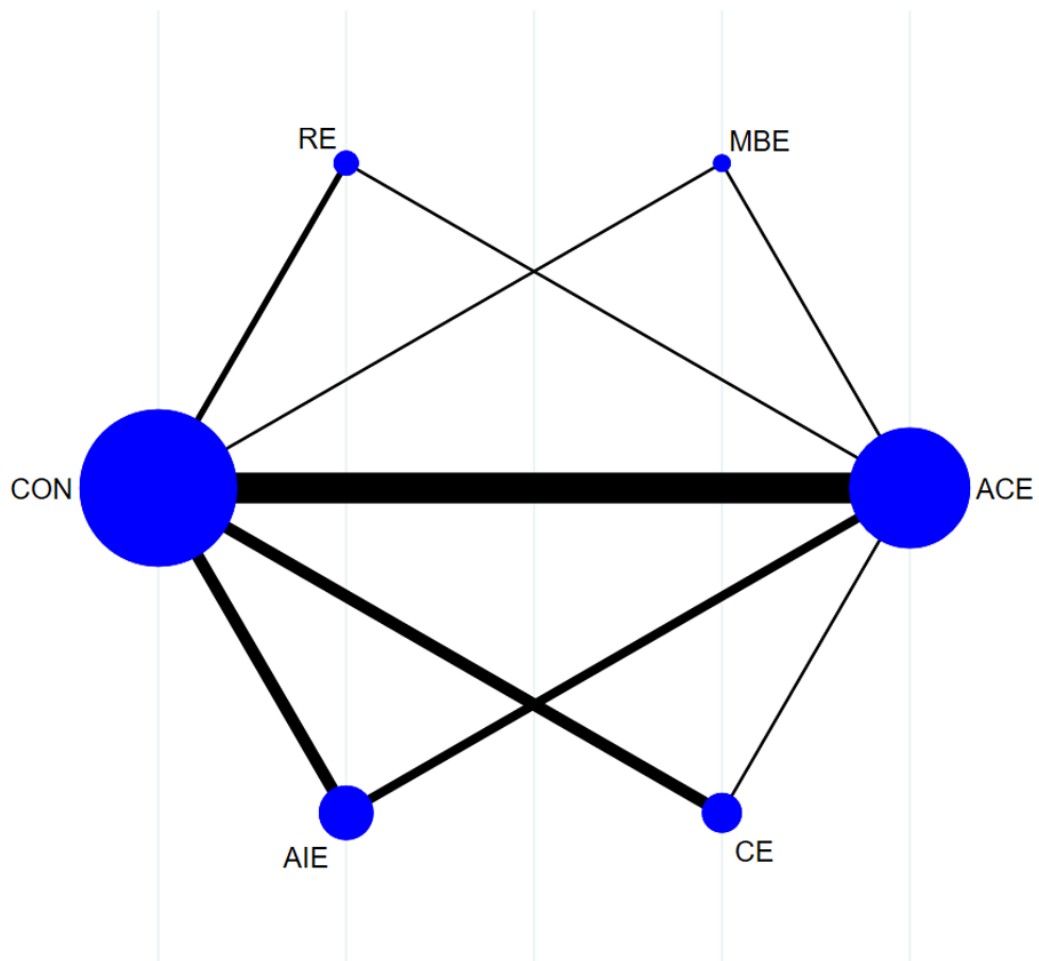

**Figure 2  Reticular evidence diagram.**

significant differences between direct and indirect evidence ($P > 0.05$), supporting the reliability of the results.

Forest plots for the eligible comparisons of brachial artery FMD, along with their 95% CI, are provided in Appendix S7. Seventeen studies assessed brachial artery FMD ($n = 797$) (*Abdi, Tadibi & Sheikholeslami-Vatani, 2021*; *Afousi et al., 2018*; *Choi et al., 2012*; *Cox et al., 2024*; *Davoodi et al., 2022*; *Desch et al., 2010*; *Gainey et al., 2016*; *Gibbs et al., 2012*; *Ku et al., 2009*; *Kwon et al., 2011*; *Li, 2009*; *Liu, Li & Liu, 2007*; *Mitranun et al., 2014*; *Okada et al., 2010*; *Ploydang et al., 2023*; *Rech et al., 2019*; *Wycherley et al., 2008*). The pooled estimates from the NMA of brachial artery FMD are presented in Table 2. AIE (MD = 2.23%, 95% CI [1.09%–3.37%], MBE (MD = 1.97%, 95% CI [0.60%–3.33%], $P < 0.05$), CE (MD = 1.17%, 95% CI [0.13%–2.21%], $P < 0.05$), and ACE (MD = 1.20%, 95% CI [0.52%–1.87%], $P < 0.05$), $P < 0.05$) all demonstrated significant improvements in brachial artery FMD compared to the control group. Table 3 and Appendix S8 present the SUCRA values for all interventions, showing that AIE (SUCRA = 89.0), MBE (SUCRA

**Table 2   Network meta-analysis matrix of results.**

| | | | | | |
|---|---|---|---|---|---|
| AIE | −0.26 (−1.99, 1.47) | −1.06 (−2.56, 0.44) | −1.03 (−2.16, 0.09) | −2.04 (−4.29, 0.20) | −2.23 (−3.37, −1.09) |
| 0.26 (−1.47, 1.99) | MBE | −0.80 (−2.50, 0.90) | −0.77 (−2.17, 0.63) | −1.78 (−4.15, 0.58) | −1.97 (−3.33, −0.60) |
| 1.06 (−0.44, 2.56) | 0.80 (−0.90, 2.50) | CE | 0.03 (−1.14, 1.20) | −0.98 (−3.20, 1.23) | −1.17 (−2.21, −0.13) |
| 1.03 (−0.09, 2.16) | 0.77 (−0.63, 2.17) | −0.03 (−1.20, 1.14) | ACE | −1.01 (−3.01, 0.99) | −1.20 (−1.87, −0.52) |
| 2.04 (−0.20, 4.29) | 1.78 (−0.58, 4.15) | 0.98 (−1.23, 3.20) | 1.01 (−0.99, 3.01) | RE | −0.18 (−2.15, 1.78) |
| 2.23 (1.09, 3.37) | 1.97 (0.60, 3.33) | 1.17 (0.13, 2.21) | 1.20 (0.52, 1.87) | 0.18 (−1.78, 2.15) | CON |

**Table 3   Ranking of exercise interventions in order of effectiveness.**

| Treatment | AIE | MBE | CE | ACE | RE | CON |
|---|---|---|---|---|---|---|
| SUCRA (%) | 89.0 | 80.1 | 51.0 | 50.9 | 20.1 | 9.2 |

= 80.1), CE (SUCRA = 51.0), ACE (SUCRA = 50.9), and RE (SUCRA = 20.1) were all more effective than the control group (SUCRA = 9.2) in improving brachial artery FMD in individuals with impaired glycaemic control.

Appendix S9 presents the funnel plots for brachial artery FMD, which were used to assess potential publication bias and small-sample effects in the NMA. The results showed that a limited number of data points displayed scattered distributions, which may be attributed to small sample sizes or other underlying biases. Egger's test further indicated significant publication bias ($P = 0.0045$). In sensitivity analysis, after excluding (*Wycherley et al., 2008*; *Desch et al., 2010*), the result of Egger's test was no longer significant ($P = 0.0726$). The SUCRA rankings were as follows: AIE (88.9), MBE (88.1), ACE (54.5), CE (41.9), RE (21.3), and CON (6.4), indicating the robustness of the NMA results.

## DISCUSSION

This study employed a systematic review and NMA to investigate the effects of different types of exercise on vascular endothelial function in individuals with abnormal glucose regulation (*i.e.*, prediabetes and T2DM). The results indicated that ACE, AIE, CE, and MBE all significantly improved brachial artery FMD, while RE demonstrated relatively weaker effects.

Firstly, this study found that AIE had the most pronounced effect on improving brachial artery FMD (SUCRA = 89.0), consistent with recent trends in related studies (*Qiu et al., 2024*; *Ramos et al., 2015*). Compared to ACE, AIE involves higher exercise intensity, which significantly increases blood flow and shear stress (*Thijssen et al., 2009*). Elevated shear stress can further activate potassium channels in vascular endothelial cells, promoting calcium ion influx and subsequently activating endothelial nitric oxide synthase, thereby enhancing nitric oxide (NO) production (*Gerhold & Schwartz, 2016*; *Ghimire et al., 2019*). NO is a crucial vasodilator that activates soluble guanylate cyclase and increases cyclic guanosine monophosphate levels, leading to relaxation of vascular smooth muscle cells and ultimately improving endothelial function (*Evora et al., 2012*). Additionally, *Ferentinos et al. (2022)* demonstrated that AIE is superior to ACE in promoting endothelial progenitor

cell (EPC) mobilization. EPCs are involved in vascular repair and endothelial function maintenance, and their increased numbers may enhance NO bioavailability, further improving endothelial function (*Zhang et al., 2021*).

Secondly, MBE (SUCRA = 80.1) ranked second in this study, indicating that mind-body exercises such as Tai Chi and Baduanjin also positively impact endothelial function. This effect may be related to their ability to effectively reduce sympathetic nervous system activity (*Cai et al., 2021*). Excessive sympathetic activity increases cardiovascular load and leads to endothelial dysfunction (*Olex, Newberg & Figueredo, 2013*), while MBE may indirectly improve endothelial health by modulating autonomic balance. Furthermore, existing studies have confirmed that MBE can improve endothelial function through multiple pathways, including lipid metabolism regulation (*Wang et al., 2024*), reduced oxidative stress, and inhibition of inflammatory responses (*Kasim, Van Zanten & Aldred, 2020*). Notably, despite its relatively low exercise intensity, MBE demonstrated significant improvements in this study, suggesting its suitability for patients with poor physical fitness or comorbid cardiovascular conditions. Future research should further explore the specific mechanisms of MBE, particularly at the molecular and cellular levels, and investigate its synergistic effects with other exercise types (*e.g.*, aerobic or resistance training) to optimize exercise interventions for diabetic patients and provide a scientific basis for individualized treatment.

Moreover, CE (SUCRA = 51.0) and ACE (SUCRA = 50.9) also demonstrated significant effects in improving brachial artery FMD, underscoring the continued relevance of traditional aerobic exercise interventions for enhancing endothelial function. CE, which combines the benefits of both aerobic and resistance training, improves cardiorespiratory endurance, muscle mass, and metabolic health (*Schroeder et al., 2019*). Studies have supported the use of CE in T2DM, suggesting that it can enhance endothelial function by promoting NO production, improving insulin sensitivity, and reducing chronic inflammation (*Kadoglou et al., 2013*; *Magalhaes et al., 2019*). The results of this study further validate these findings, suggesting that CE may provide a comprehensive and suitable intervention for a broad range of patients.

In contrast, RE (SUCRA = 20.1) demonstrated a relatively weaker effect on brachial artery FMD. This result is consistent with several previous meta-analyses (*Cortes et al., 2023*; *Da Silva et al., 2024*). While RE is known to increase skeletal muscle mass and improve glucose and lipid metabolism (*Cauza et al., 2005*), its direct impact on endothelial function remains a subject of debate. For instance, studies by *Rech et al. (2019)* and *Kwon et al. (2011)* found that a 12-week RE program did not result in significant improvements in brachial artery FMD in T2DM patients, whereas *Cohen et al. (2008)* reported that a longer RE program (14 months) significantly improved endothelial function. This suggests that the duration of exercise may be a key determinant of RE's effects on endothelial health. Additionally, training parameters such as load, frequency, and number of sets may influence the vascular endothelial effects of RE. Consequently, future research should investigate the impact of various RE protocols on endothelial health, particularly within diabetic populations, to refine exercise intervention strategies and provide more precise clinical recommendations.

## LIMITATIONS

Firstly, some included studies had a certain risk of bias, particularly in missing data (D3) and selective reporting (D5), which may affect the reliability of the results. Secondly, there was some heterogeneity in the intervention protocols (*e.g.*, exercise frequency, duration) across studies, which may influence the degree of brachial artery FMD improvement. Additionally, the limited number of studies on MBE, RE, and CE may restrict the generalizability and representativeness of this study's findings.

## CONCLUSION

This NMA indicates that AIE is the most effective in improving brachial artery FMD, followed by MBE, which is suitable for patients with poor physical fitness or comorbid cardiovascular conditions. CE and ACE also effectively improve brachial artery FMD, supporting their clinical application. In contrast, RE showed weaker effects, which may be influenced by training duration and program design. Future research should further explore the long-term effects and individualized intervention strategies of different exercise modalities to optimize exercise therapy for prediabetes and T2DM populations.

### Funding
The authors received no funding for this work.

### Competing Interests
The authors declare there are no competing interests.

### Author Contributions
- Zongxiang Li conceived and designed the experiments, performed the experiments, analyzed the data, prepared figures and/or tables, authored or reviewed drafts of the article, and approved the final draft.
- Shengyao Luo performed the experiments, authored or reviewed drafts of the article, and approved the final draft.
- Xuebing Bai performed the experiments, prepared figures and/or tables, and approved the final draft.
- Lu Huang conceived and designed the experiments, prepared figures and/or tables, authored or reviewed drafts of the article, and approved the final draft.
- Hongyan Guo analyzed the data, authored or reviewed drafts of the article, and approved the final draft.
- Song Chen analyzed the data, prepared figures and/or tables, and approved the final draft.
- Dan Wang analyzed the data, authored or reviewed drafts of the article, and approved the final draft.

## Data Availability

  The raw measurements are available in the Supplemental File.

## Supplemental Information

Supplemental information for this article can be found online at http://dx.doi.org/10.7717/peerj.19839#supplemental-information.

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
