# Peer review of "Effects of different exercise types on vascular endothelial function in individuals with abnormal glycaemic control: a systematic review and network meta-analysis"

_PeerJ, doi:10.7717/peerj.19839_

## Round 0.1 · original submission · Major Revisions

·

Basic reporting

The results of the risk of bias, forest plot, funnel plot, and data extraction table are mentioned as available in the appendix, but I cannot find them in the supplementary file provided.

Experimental design

No comments.

Validity of the findings

No comments.

Additional comments

I have a query regarding the inclusion criteria for the population, which states adults aged 18 years and older. The physiology of individuals aged 65 years and above changes. Considering these age-related physiological changes, how do you generalize the findings of your review? I would suggest that a subgroup analysis based on age would provide us with more clarity and generalizability.

Reviewer 2 ·

Basic reporting

Line 35: The authors should specify which guideline or technique was used to assess the risk of bias.

Lines 37–40: The results related to flow-mediated dilation (FMD) are missing the unit of measurement. Please provide the appropriate unit, especially since the outcomes are reported as mean differences.

Lines 82–83: The transition from introducing FMD to discussing exercise, diet, and other interventions lacks clarity. Before focusing on exercise, it would be more logical to first discuss other approaches used to assess or improve FMD, such as statin therapy, particularly in patients with type 2 diabetes mellitus (T2DM). Please consider citing and discussing the following studies: DOI: 10.1097/MD.0000000000032313; DOI:10.1016/j.atherosclerosis.2012.01.031
Highlight the limitations of these approaches to justify the focus on exercise as the intervention of interest.

Line 119: Ensure that the reference is placed before the full stop.

Lines 140 and 145: Similarly, references should be placed before the full stop.

Line 135: Please provide a reference supporting the PICOS criteria you adopted.

Line 146: Define all acronyms (ACE, AIE, RE, MBE, and CE), as this is the first time they appear in the main text.

Line 153: Define RCT in full as "Randomized Controlled Trial" at first mention.

Line 176: Insert the reference before the full stop.

Lines 185–186: The formula used to estimate the change in standard deviation is unclear. It would be helpful to include the correlation matrix that was used in the calculation.

Experimental design

The study is registered with PROSRERO which is well commendable, as this allows transparency
The control group is not clearly defined. Please specify the exact control condition used in your study.
Line 191: Evaluating publication bias using only statistical tests, such as Egger’s test, may be insufficient for drawing robust conclusions. It is recommended to also include a funnel plot for visual inspection. Therefore, lines 212–213 should be integrated with the content in line 191 to improve the flow and clarity of this section.

Validity of the findings

Line 191: Evaluating publication bias using only statistical tests, such as Egger’s test, may be insufficient for drawing robust conclusions. It is recommended to also include a funnel plot for visual inspection. Therefore, lines 212–213 should be integrated with the content in line 191 to improve the flow and clarity of this section. Currently, it is not clear the value of r that was used in the calculation of SD.

---

## Round 0.2 · accepted · Accept

The authors have addressed the issues/concerns in the manuscript. I recommend the manuscript for publication. Thanks,

Reviewer 2 ·

Basic reporting

Comments adequately addressed

Experimental design

No comments; previous comments fully addressed

Validity of the findings

Comments fully adressed